# Mammary Paget’s Disease: An Update

**DOI:** 10.3390/cancers14102422

**Published:** 2022-05-13

**Authors:** Sione Markarian, Dennis R. Holmes

**Affiliations:** Department of Surgery, Adventist Health Glendale, Glendale, CA 91206, USA; sione@drholmesmd.com

**Keywords:** mammary Paget’s disease, MPD, breast malignancy, cryoablation, Toker cells

## Abstract

**Simple Summary:**

Mammary Paget’s disease of the breast is an uncommon and often misdiagnosed breast malignancy. The review discusses the diagnosis, work-up, treatment, and prognosis of mammary Paget’s disease.

**Abstract:**

Mammary Paget’s Disease is a non-invasive cutaneous malignancy of the breast involving the nipple-areolar complex that is commonly mistaken for benign breast conditions, leading to delay in diagnosis. This review article discusses Paget’s disease etiology, clinical presentation, differential diagnosis, diagnostic work-up, natural history and prognosis. This article also discusses evolving strategies for the surgical and non-surgical management of Paget’s disease.

## 1. Introduction

Paget’s disease of the breast, or mammary Paget’s disease (MPD), was first described by Sir Paget in 1874 as an eczematous lesion of the nipple associated with an underlying cancer [1]. Now, MPD is recognized as a rare cutaneous intraepithelial malignancy characterized by large epidermal adenocarcinoma cells, called Paget’s cells, within the squamous epithelium of the nipple, which may extend into the areola and adjacent skin [2]. Paget’s disease may also develop on ectopic breasts and accessory nipples [3]. Histopathologically, Paget cells are characterized by the presence of clear, abundant cytoplasm and enlarged, pleomorphic hyperchromatic nuclei that are often found in a basal layer without any intercellular bridges to adjacent cells [4]. These cells appear organized in small clusters or large sheaths that may sometimes completely replace the epidermal cells. Paget cells have similar immunohistochemical staining patterns as underlying breast cancer cells, including expression of carcinoembryonic antigen, epithelial membrane antigen, and some mucins [5]. Similar to the underlying carcinoma cells, Paget cells demonstrate *HER2* oncogene positivity, but estrogen and progesterone antigens are frequently negative [4,5].

Two main theories have been proposed for how MPD arises: the epidermotropic theory and the intraepidermal origin theory. The epidermotropic theory states that Paget cells originate from underlying intraductal carcinoma cells that migrated along the basement membrane into the nipple. This theory is supported by the high incidence of MPD with underlying ductal carcinoma in situ (DCIS) [5]. Further evidence in support of the epidermotropic theory is the high level of *HER2/neu* oncogene expression in Paget cells, which mirrors the level of *HER2/neu* expression in underlying DCIS [1]. The intraepidermal origin theory states that Paget cells result from malignant transformation of pluripotent keratinocyte stem cells or cells of apocrine gland ducts in the absence of underlying carcinoma, which also provides an explanation for cases of MPD where the nipple is spared. This theory is supported by the morphological similarity between Paget cells and Toker cells, which are non-malignant, cytoplasm-rich epithelial cells of sebaceous-gland origin detected in the areolar skin of 10% of women [6]. However, unlike Paget cells, which generally strongly express HER2/neu and ki-67, Toker cells are uniformly HER2/neu negative and express very low levels of ki-67. A hybrid theory proposes that Paget cells can originate either epidermotrophically or intraepidermally, depending on the circumstances and local conditions [5].

## 2. Presentation

Mammary Paget’s disease is reported in 1–3% of all primary breast cancers [7]. Between 93–100% of MPD cases are associated with underlying breast cancer, commonly central and multifocal tumors, mainly located near the areola [5]. MPD can be divided into three different categories based on the presence or extent of associated disease: (1) MPD of the nipple without DCIS, (2) MPD of the nipple with DCIS in the underlying lactiferous ducts within 2 cm of the nipple, and (3) MPD of the nipple with DCIS in the underlying lactiferous ducts and associated DCIS or invasive breast cancer elsewhere in the breast extending ≥2 cm from the nipple-areolar complex [8]. More than 90% of cases are associated with underlying DCIS or invasive ductal carcinoma (IDC) [7]. In a study conducted at the M.D. Anderson Cancer Center, of the 104 patients studied, 63 (60%) had invasive carcinoma, and 34 patients (33%) had DCIS [9]. Only 7 patients (7%) had Paget’s disease of the nipple, where the lesion was confined within the nipple without evidence of invasive or noninvasive disease [9].

In a study of 70 women with a clinical diagnosis of MPD conducted between 1971 and 1999 by Kothari, et al., 60% of all cases of underlying invasive tumors were grade III carcinomas with increased predisposition to metastasis and poorer prognosis [10]. Additionally, 96.5% of cases of underlying DCIS were of a high nuclear grade with a greater risk of developing high-grade invasive disease [10]. A high incidence of HER2/neu-positive carcinomas is one factor associated with higher nuclear grade and worse overall prognosis for patients with MPD associated with invasive carcinoma.

## 3. Clinical Features

Initially, MPD develops insidiously, gradually evolving over months to years as it extends from the nipple into the areola in a centrifugal growth pattern [1]. Typically, MPD presents clinically as a unilateral rash of the nipple and areola that in more advanced cases may also involve the periareolar skin. Skin rashes can range up to 15 cm in diameter. MPD skin changes of the nipple and/or areola may resemble eczema with a fine scaling erythematous rash or a flaky, fissured, bleeding rash in more established cases. Advanced cases are often accompanied by skin ulceration and nipple retraction. Hyperpigmented lesions similar to superficial spreading melanoma have also been described [5]. MPD can affect male patients who present with similar clinical characteristics as those occurring in women. However, the prognosis seems to be worse in men compared to women [3,5]. Figure 1 displays multiple examples of MPD. Figure 2 displays the 2-year progression of MPD as documented by an individual patient. Table 1 presents the frequencies of presenting signs and symptoms of MPD from a population-based cohort study of 223 women with histologically verified MPD of the nipple diagnosed between 1976 and 2001 at 13 Swedish hospitals [9]. In 98% (217) of the patients, the main presenting symptom was eczema or ulceration of the nipple [9]. In the early stages, however, the nipple appears to be normal, but the patient might present with mild symptoms such as nipple pruritus. MPD-associated skin surface changes slowly progress, producing a dermatitictous appearance affecting the nipple, areola, and eventually the skin of the breast [8]. Commonly, MPD is initially misdiagnosed as eczema, dermatitis, or psoriasis, which accounts for the frequent delay in diagnosis. Consequently, to avoid a delayed diagnosis, any suspected signs or symptoms of eczematoid, pigmented, crusted, or scaly lesions or chronic inflammation in the nipple should be confirmed with biopsy [4]. Patients with misdiagnosed MPD often receive extended courses of a topical treatment without significant improvement, which further delays diagnosis, although transient responses may be observed at the margins of the skin lesion [8]. Thus, awareness and detailed physical examination are pivotal to distinguishing MPD from other benign (e.g., psoriasis, dermatitis, chronic eczema, lactiferous duct ectasia, or adenomatosis of the nipple) or malignant conditions (e.g., cutaneous extension of breast carcinoma, Bowen’s disease, basal cell carcinoma, or melanoma) involving the nipple-areolar complex [5]. Bowen’s disease (squamous cell carcinoma in situ) and MPD are both malignant processes that cause changes in the nipple-areolar complex skin; however, Bowen’s disease cannot be clinically differentiated from MPD without a biopsy [8]. Table 2 compares common signs and symptoms of MPD with other conditions that may affect the nipple-areolar complex [4,11].

## 4. Work-Up

MDP is diagnosed initially based on clinical presentation, physical examination, and breast imaging. Both benign and malignant processes can produce visible symptoms in the skin of the nipple. However, if apparently benign skin changes do not improve after a two-week course of topical corticosteroids, a diagnostic imaging work-up and biopsy should be performed.

Efficient work-up includes high-quality diagnostic imaging to rule out malignancy due to the high probability of breast carcinoma associated with MPD. MPD co-exists with ductal carcinoma in situ in more than 93% of cases, while fewer than 10% of cases are associated with a palpable mass. Multifocality and multicentricity are reported in 41 and 34% of MPD cases, respectively [8]. In the study by Kothari, et al., all patients with MPD presenting with a palpable mass had multifocal disease, and 30% had multicentric disease [10]. Among those who did not have a mass at presentation, 63% had multifocal or multicentric diseases [10].

Mammography should be used as the primary diagnostic imaging modality for detecting underlying carcinoma, followed by breast ultrasound if the mammogram is negative. Mammographic findings may include skin thickening of the nipple-areolar region, asymmetric density, nipple retraction, a discrete mass, and/or suspicious microcalcifications [14]. Mammography is 97% sensitive in detecting an underlying malignancy in MPD cases if a palpable mass is present clinically; however, it only detects underlying malignancy in 50% of cases if no palpable mass is present [2]. Following their study of 48 women with MPD, Dixon, et al. recommended against using mammography alone to evaluate MPD since it failed to reveal evidence of underlying disease in 43% of patients with histologically confirmed carcinoma [15]. Table 3 summarizes the mammographic findings of a retrospective study that reviewed the clinical, pathologic, and mammographic records of 58 patients with biopsy-proven MPD [16].

Breast ultrasound findings may include a mass, microcalcifications, ductal ectasia, and/or flattening, asymmetry, and thickening of the nipple-areolar complex [8]. Ultrasound may also be utilized to assess the appearance of the axillary nodes.

Contrast-enhanced breast magnetic resonance imaging (MRI) may also be used to assess the extent of disease in a patient with positive mammogram or ultrasound findings, especially when breast-conserving surgery is being contemplated. Among patients with suspected or confirmed MPD, contrast-enhanced breast MRI can also be useful in detecting occult, multifocal, or multicentric lesions when there is no clinical sign or significant mammographic or ultrasound findings [8]. By MRI, the abnormal nipple-areolar complex may be characterized by asymmetric nodular, discoid, or irregular enhancement compared to the unaffected, contralateral breast [8].

In a retrospective study of 58 patients, Siponem, et al. found mammography to be 79% sensitive and ultrasound to be 74% sensitive in detecting invasive cancer; however, both studies were less sensitive at detecting DCIS [mammography (39%) ultrasound (19%)] [15]. In the same study, MRI was 100% sensitive for infiltrating carcinoma and 44% sensitive for DCIS [15].

Whether or not underlying malignancy is identified, patients presenting with a persistent nipple-areolar rash and suspected MPD should undergo a full-thickness biopsy of the nipple or areolar using a 2–4 mm diameter punch biopsy tool or a full-thickness incisional biopsy [4]. When an adequate perch cannot be obtained on the nipple papilla to permit a skin punch biopsy, the surgeon may instead perform a wedge-shaped full-thickness incisional biopsy of the nipple papilla. Incisional biopsy incisions and larger diameter skin punch biopsy wounds should be closed with non-absorbable sutures due to the increased risk of delayed wound healing. Surgical specimens should be submitted to pathology in formalin for histological assessment, along with a description of the clinical presentation and clinical impression. 

## 5. Treatment

There are no category 1 data that specifically address the local management of MPD according to the 3.2022 version of the National Comprehensive Cancer Network guidelines (accessed on 23 March 2022). Therefore, surgical management remains the main intervention for MPD based on the management of non-MPD-associated breast cancer. Total or skin-sparing mastectomy with surgical axillary staging with or without breast reconstruction is frequently performed for the treatment of MPD due to the frequency of multicentric or multifocal disease [14]. However, with improvements in imaging and patient selection, breast-conserving therapy has become increasingly more common for patients with unifocal disease limited to the nipple-areolar region. Removal of the nipple-areolar complex is generally achieved with a central lumpectomy for removal of the involved nipple and areola en bloc with underlying disease (Figure 3). Superior esthetic results can be achieved with oncoplastic surgical techniques (e.g., Grisotti mastopexy, Wise-Pattern mammaplasty) combined with a contralateral mammaplasty or mastopexy, if desired, to maintain breast symmetry (Figure 4). For optimal esthetic results, immediate or delayed nipple-areolar reconstruction (e.g., C-V Flap) and/or dermatography (medical tattooing) may be performed to create a symmetrical, color-matched nipple-areolar complex following resection of the nipple and/or areola.

Non-surgical or limited surgical approaches have also been utilized to manage MPD. In cases without evidence of underlying disease, non-operative management can offer an effective alternative therapy to traditional breast-conserving therapy. In a study by Bulens, et al., 13 patients with MPD confined to the nipple or surrounding skin without signs of an underlying tumor were treated with radiotherapy alone without surgical resection, with no recurrences detected after 58.6 months of mean follow-up [7]. Alternatively, local excision without radiotherapy may be utilized in MPD cases limited to the skin. Investigational therapies such as photodynamic therapy (PDT) have been used to treat cases of MPD as a less invasive alternative therapy. In PDT, a topical or intravenous photosensitizer drug is administered, and a specific wavelength of light is used to activate the drug, which binds with oxygen to destroy the affected cells. Studies on the non-operative management of PDT are limited, and more research is necessary to determine its safety and effectiveness [17]. The authors have also employed cryoablation of subareolar lesions combined with local excision of the affected nipple and/or areola to manage MPD and carcinoma limited to the skin and subareolar tissue among patients refusing a formal partial mastectomy (Figure 5).

## 6. Prognosis

Nearly all patients diagnosed with MPD have either underlying invasive or intraductal carcinoma [18]. In patients with MPD and no palpable or mammographic mass, the majority will have underlying DCIS; therefore, axillary lymph nodes are usually negative, and treatment should be limited to the breast [18]. In a meta-analysis of mastectomy patients, a breast carcinoma-specific mortality rate of 1.7% was reported at 8.6 years of follow-up. Similarly, mastectomy for MPD has also resulted in high rates of local control and survival, although some studies have reported that invasive recurrences have occurred following surgery, indicating that mastectomy does not necessarily result in a 100% cure rate [18]. Conservative management of DCIS using BCS has been highly successful and has demonstrated significant reduction in the risk of local recurrence and invasive recurrence in patients receiving whole breast radiotherapy following lumpectomy [18].

The study by Kothari, et al. [10] of 70 patients with MPD found that cone excision of the nipple results in incomplete excision 75% of the time due to MPD’s association with underlying malignancy. High recurrence of local excision was also found in a study by Dixon, et al. with a 40% recurrence rate in patients with an in situ component in proximity to the nipple [19]. If DCIS or invasive cancer is located further from the nipple-areolar region, complete resection of the area and nipple-areolar complex is recommended, en bloc or individually, followed by radiotherapy, assuming negative margins are obtained. Sentinel node biopsy or axillary node dissection may be performed based on the extent of lymph node involvement.

In a retrospective study of 200 women with MPD followed for 25 years, Dalberg, et al. studied the effects of various treatments on survival and recurrence [15]. Twenty percent of patients were treated with local surgery, and the rest had a total mastectomy. They concluded that the type of surgery performed had no influence on breast cancer or disease-free survival; thus, breast-conserving surgery may have similar results to total mastectomy. The only two risk factors they found for recurrence or death were underlying invasive cancer and the presence of a palpable mass.

Another prospective study observing local recurrence following breast-conserving therapy and radiotherapy included 61 patients with MPD without underlying invasive carcinoma [15]. Of patients with DCIS, 93% were treated with cone excision of the nipple-areolar complex and underlying breast tissue, which was followed by whole-breast radiation. The same treatment was applied to 7% of the patients who only had MPD of the nipple [15]. After a median follow-up of 75 months, only 7% of the 61 patients had local recurrence [15].

In a study by Dubar, et al., 36 patients with MPD and no underlying palpable mass or mammographic anomaly underwent complete or partial resection of the areola and radiotherapy, of which 11% developed a local recurrence at a median follow-up of 112 months [1]. Local recurrence was detected in 14% (3/22) of patients who had complete resection of the nipple-areolar plaque combined with whole breast radiotherapy followed by a tumor bed boost [1]. Of the six patients who had a partial resection of the nipple-areola complex and whole-breast radiotherapy plus boost, two developed local recurrences, and one of these had both local and distant recurrence [1]. The four women with isolated local recurrences were successfully managed with mastectomy and remained disease-free at the last follow-up median of 112 months [1].

Additionally, a 2003 report by Marshall, et al. for the American Cancer Society studied 38 cases of biopsy-proven MPD from 7 institutions between 1980 and 2000 treated with BCS and radiotherapy with a median follow-up of 113 months [18]. At the time of diagnosis, a pathological review revealed typical Paget cells of the nipple in 36 cases, 30 (83%) of which had an underlying malignancy [18]. A total of 4 of the 36 patients (11%) who had undergone complete resection of the nipple-areolar complex at primary surgery developed local disease recurrence 12–69 months after completion of radiotherapy [18]. DCIS was only present in two of these four patients [18]. Two additional patients (6%) who had partial nipple-areolar complex excision at primary surgery developed an in-breast recurrence simultaneously with either regional or distant recurrence of invasive and intraductal disease in both cases [18]. At the most recent follow-up visit, local disease was controlled for 35 of the 36 patients (97%) either by primary treatment (30 patients) or salvage surgery (5 patients) [18]. The European Organization for Research and Treatment of Cancer found that in a cohort of 61 patients with MPD treated with BCS and radiotherapy, the 5-year local recurrence rate was 5.2% compared with the 9% 5-year local recurrence rate in the American Cancer Society study [18].

Without treatment, the skin lesions and underlying disease will progress to the development of invasive breast cancer, potentially followed by lymph node and visceral metastasis. The presence of a palpable breast tumor, lymph node enlargement, high nuclear grade, and age below 60 years are unfavorable prognostic factors that correlate with a high risk of invasive carcinoma and a high rate of lymph node metastasis [3].

Many studies have found that MPD negatively influences breast cancer survival, consistent with its tendency to develop in association with higher-stage disease. Ordz-Pagan conducted a study comparing MPD and non-MPD groups and found that the non-MPD group had an overall 5-year survival rate of 93.8% compared to 81.2% for MPD; however, there was no difference in disease-free survival (DFS) [20]. However, when controlling for HER2/neu status, age, tumor size, nuclear grade, and nodal status, Kothari, et al. reported that individual non-MPD case-controls experienced almost identical overall survival rates as patients with MPD, although the length of follow-up was not specified [10].

Lymph node status correlates with overall survival with 75–95% 5-year overall survival in patients with negative lymph nodes compared to only 20–25% in patients with positive lymph nodes [3]. There is no evidence that MPD behaves differently in males; however, their 5-year survival rate is lower compared to women (20–30% in males versus 30–50% in females), consistent with a pattern of higher stage at presentation among male breast cancers [21]. In 2012, the International Agency for Research on Cancer reported that the 5-year recurrence-free survival was 75–90% for those with DCIS and 63–75% for those with invasive carcinoma [22]. The 5-year overall survival rates are 94–98% in the presence of DCIS and 73–93% in cases with invasive carcinoma, depending on the stage of presentation and tumor biology [2,22]. However, the relationship between tumor genomics and survival has yet to be characterized for MPD.

## Figures and Tables

**Figure 1 cancers-14-02422-f001:**
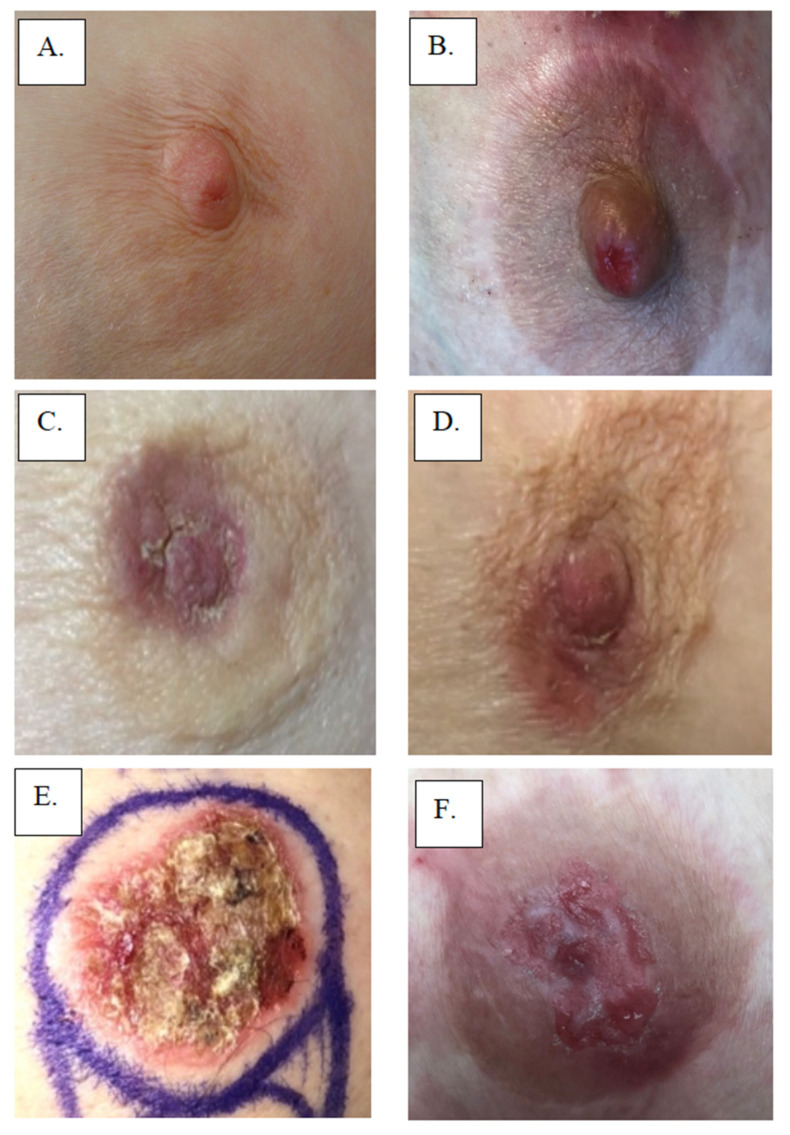
Examples of Paget’s Disease: (**A**,**B**) involving nipple only, (**C**,**D**) involving nipple and areola, and (**E**,**F**) showing effacement of the nipple.

**Figure 2 cancers-14-02422-f002:**
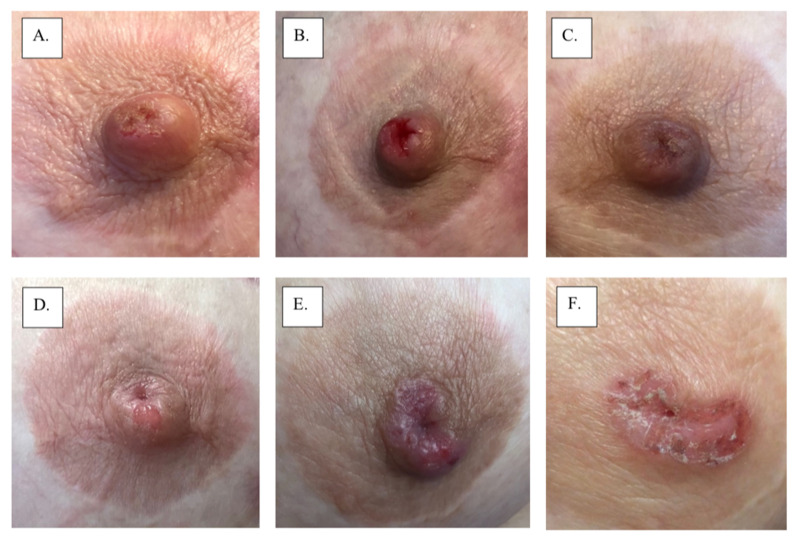
Images A-F show progression of mammary Paget’s disease over a 2-year period.

**Figure 3 cancers-14-02422-f003:**
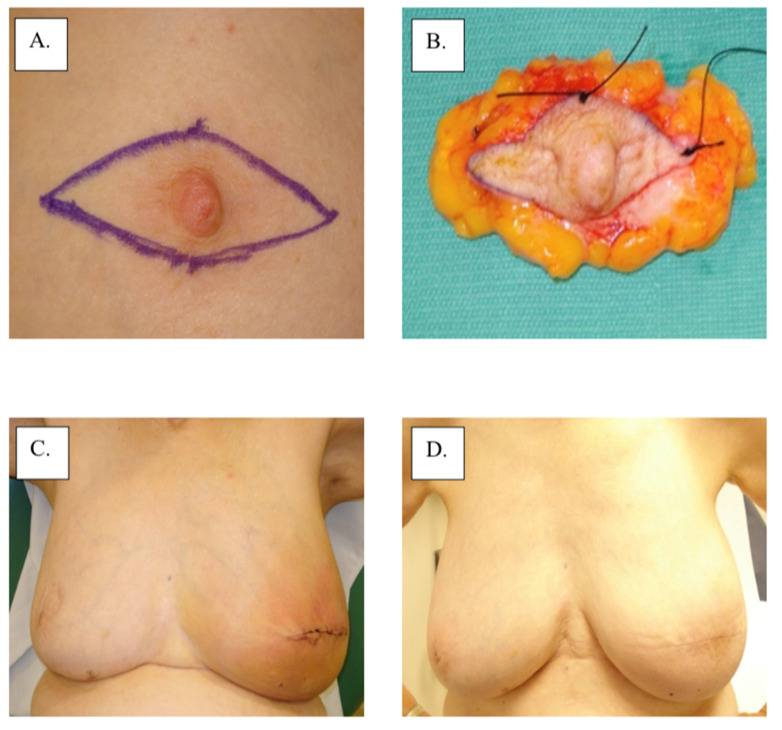
Surgical management with central lumpectomy: (**A**) surgical plan, (**B**) surgical specimen, (**C**) immediate post-operation, and (**D**) 1-year post-operation.

**Figure 4 cancers-14-02422-f004:**
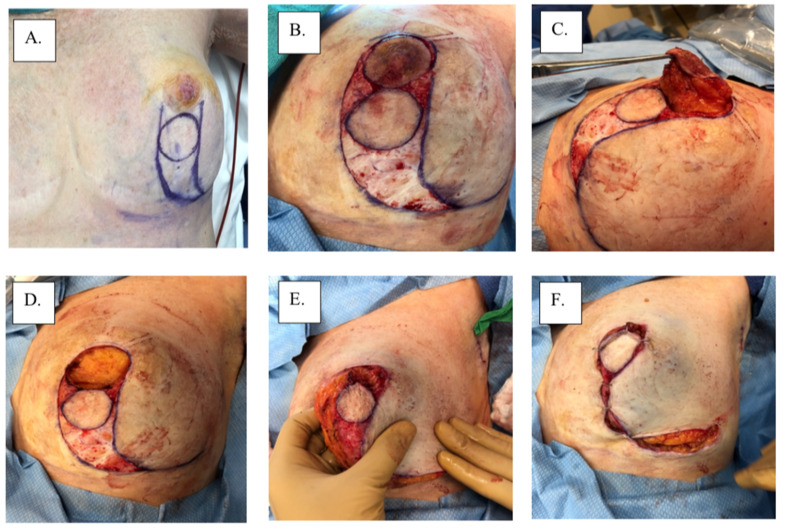
Surgical management with Grisotti procedure: (**A**) surgical plan, (**B**) de-epithelized skin, (**C**) surgical specimen, (**D**) surgical cavity, (**E**) superior-medial rotation of lower outer quadrant, (**F**) tailor-tacked skin edges, (**G**) initial wound closure, (**H**) immediate post-operation, and (**I**) 1-year post-operation.

**Figure 5 cancers-14-02422-f005:**
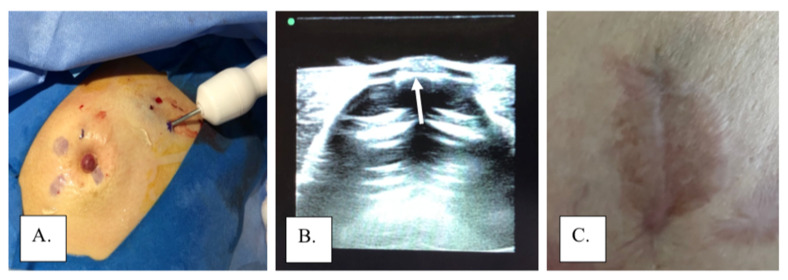
Cryoablation of subareolar lesion with local excision of excision: (**A**) cryoprobe during ablation of a subareolar lesion, (**B**) ice ball extending to the base of the nipple (arrow) viewed through a gel standoff pad, and (**C**) areola following excision of nipple under local anesthesia.

**Table 1 cancers-14-02422-t001:** Presenting symptoms of MPD.

Presenting Symptoms and Signs among 223 Patients [9]	Percentage of Patients Displaying Each Symptom
Eczema or ulceration of the nipple	98%
Malignancy suspicious mammogram	32%
Palpable breast mass	15%
Bloody nipple discharge	10%

**Table 2 cancers-14-02422-t002:** Comparison between MPD and conditions affecting the nipple-areolar complex.

Differential Diagnosis of Paget’s Disease [4,11,12,13]	Features	
	**Other Conditions**	**Paget’s Disease**
Eczema	May be bilateral More common premenopausal Nipple is usually intact No underlying lump Itchy Responsive to steroids	Unilateral More common postmenopausal Nipple is usually distorted Underlying lump may be present Not itchy or slightly itchy Non-responsive to steroids
Psoriasis	Vesicles and pustules	No vesicles and pustules
Irritant contact dermatitis	No change in the nipple	Nipple retraction or deformation
	Limited to areola	Involves nipple, may extend to areola
Mammary duct ectasia	Usually bilateral	Usually unilateral
Drug eruption	No palpable mass	Palpable mass may be present
Toker cells	Common in younger age	Common in older age
Nipple duct adenoma	Normal mammograms	Mammograms frequently abnormal
Bowen’s Disease	The presence of intercellular bridges favors Bowen’s disease. The skin of the nipple is usually uninvolved. Bowen’s disease more commonly appears on areas of the skin that have been exposed to the sun. Major risk factors for Bowen’s disease include ultraviolet radiation, human papillomavirus infection and immunosuppression.	Glandular formation within the epidermis is more commonly seen in Paget’s disease. The clinical lesion usually starts from the nipple then extends to the areola and surrounding skin. There is no association with sun exposure or HPV.

**Table 3 cancers-14-02422-t003:** Mammographic findings in patients with biopsy-proven MPD.

Mammogram Findings among 58 Patients [16]	Percentage of Patients Exhibiting Findings
Normal findings	31%
Nipple, areolar, or subareolar abnormalities	24%
Evidence of masses or calcifications	45%

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
