# Peer review of "Mammary Paget’s Disease: An Update"

_cancers, 2022, doi:10.3390/cancers14102422_

Round 1

Reviewer 1 Report

dear author team.

thank you for submitting your work.

Your review adresses a relatively rare type of cancer as you adequately describe. therefore, attention to the diagnosis is wise.

in general I feel you adress most items in your manuscript. I do recommend looking at more clear headings of subheadings to get rid of some of the duplicate information you share, e.g. on biopsying.

perhaps; introduction/prognosis overview; presentation local - presentation with underlying malignancy- physical exam - workup (biopsy) - imaging - treatment local - treatment/staging lymph nodes - adjuvant therapy 

adding structure might make things more readible.  

more detailed;

figure 1  i would put introduce this earlier in the manuscript. a suggestion is to make a separate block on biopsy in the algorithm and then go to positive/negative

i miss a short comment about nipple reconstruction surgically and or with dermatography.

page 12 line 257; when two patients comprize 33% it is better to leave the percentages.

Author Response

in general I feel you adresss most items in your manuscript. I do recommend looking at more clear headings of subheadings to get rid of some of the duplicate information you share, e.g. on biopsying.

perhaps; introduction/prognosis overview; presentation local - presentation with underlying malignancy- physical exam - workup (biopsy) - imaging - treatment local - treatment/staging lymph nodes - adjuvant therapy 

adding structure might make things more readible.  

Based on the reviewers suggestion, the heading have been modified and ordered as follows:

Introduction
Presentation
Clinical Features
Work-up
Treatment
Prognosis

more detailed;

figure 1  i would put introduce this earlier in the manuscript. a suggestion is to make a separate block on biopsy in the algorithm and then go to positive/negative

We determined that it is best to position figure 1 in the “Presentation” section.

The algorithm has been adjusted.

i miss a short comment about nipple reconstruction surgically and or with dermatography.

The authors have been added this to the treatment section.

page 12 line 257; when two patients comprize 33% it is better to leave the percentages.

This correction has been made.

Reviewer 2 Report

Well written review of Paget's disease. Just couple of clarifications needed

  1. If nipple is spared and only areola is affected with itchy rash, can we rule out Paget's disease?  
  2. Authors advice a full thickness, wedge shaped incisional biopsy of nipple to diagnose Paget's disease. However, in many centres, punch biopsy of the nipple is used rather than Wedge biopsy. 

Author Response

  1. If nipple is spared and only areola is affected with itchy rash, can we rule out Paget's disease?  

    No, it is possible for Paget’s to arise in the areolar (and spare the nipple).  The could originate from Toker cells (i.e., pluripotential theory).

  2. Authors advice a full thickness, wedge shaped incisional biopsy of nipple to diagnose Paget's disease. However, in many centres, punch biopsy of the nipple is used rather than Wedge biopsy. Agreed.

    The option of nipple punch biopsy is added.  The ability to do so depends on the extent and location of nipple involvement, and whether or not there is sufficient perch for a skin punch biopsy.